# Hierarchical Residual Attention Network for Musical Instrument Recognition Using Scaled Multi-Spectrogram

**Rujia Chen ***, **Akbar Ghobakhlou**  **and Ajit Narayanan** 

Computer Science and Software Engineering Department, Auckland University of Technology, Auckland 1010, New Zealand; akbar.ghobakhlou@aut.ac.nz (A.G.); ajit.narayanan@aut.ac.nz (A.N.)
* Correspondence: rujia.chen@autuni.ac.nz

**Featured Application: This proposed method could potentially be applied to musical instrument classification tasks, contributing to the organization and analysis of audio data where identifying instruments is required. This may be useful in research, music information retrieval systems, or other related applications.**

**Abstract:** Musical instrument recognition is a relatively unexplored area of machine learning due to the need to analyze complex spatial–temporal audio features. Traditional methods using individual spectrograms, like STFT, Log-Mel, and MFCC, often miss the full range of features. Here, we propose a hierarchical residual attention network using a scaled combination of multiple spectrograms, including STFT, Log-Mel, MFCC, and CST features (Chroma, Spectral contrast, and Tonnetz), to create a comprehensive sound representation. This model enhances the focus on relevant spectrogram parts through attention mechanisms. Experimental results with the OpenMIC-2018 dataset show significant improvement in classification accuracy, especially with the "Magnified 1/4 Size" configuration. Future work will optimize CST feature scaling, explore advanced attention mechanisms, and apply the model to other audio tasks to assess its generalizability.

**Keywords:** spectrograms; musical instrument classification; audio classification; audio feature extraction; music information retrieval; spectrogram transformation; residual attention networks; attention mechanisms; deep learning for audio

## 1. Introduction

Musical instrument recognition in recorded music is complex due to the diverse and intricate features in audio signals. Traditional approaches using individual spectrograms like short-time Fourier Transform (STFT), logarithm Mel frequency (log-Mel), or Mel-frequency cepstral coefficients (MFCCs) capture only specific aspects of the audio signal, such as amplitude and the short-term power spectrum. This often fails to capture the full richness of audio features necessary for accurate instrument classification.

To address this, combining multiple spectrograms—such as STFT, Log-Mel, MFCC, and CST features (Chroma, Spectral contrast, and Tonnetz)—into a single compact input image has been proposed for various audio tasks [1–5]. This approach leverages the strengths of each spectrogram type, providing a more comprehensive understanding of the audio signal and leading to more accurate classification accuracy.

However, the increased number of spectrogram features can introduce challenges, making it difficult for the model to determine which part of the input image is the most important. Attention mechanisms help the model focus on the most relevant spectrogram features, addressing this issue. Some studies, like those on bird sound recognition [3] and music annotation [6], integrate attention mechanisms [7,8] to enhance the model.

*Original Contribution*

Expanding on the foundations discussed above, we here introduce a novel Hierarchical Residual Attention Network (HRAN) specifically designed for musical instrument classification. Our approach is distinct due to the integration of multi-scaled spectrograms—combining STFT, Log-Mel, MFCC, and CST (Chroma, Spectral contrast, and Tonnetz) features. This multi-scale integration enables the model to capture a broad spectrum of audio features, encompassing tonal, rhythmic, and harmonic components that are often underrepresented when using single spectrogram types.

Our HRAN framework incorporates layered attention mechanisms strategically placed within the hierarchical structure. This attention system allows the model to dynamically focus on the most relevant parts of the spectrograms, thereby improving feature discrimination and classification accuracy. By scaling each spectrogram type, we tailor feature representations for optimal learning. The model's architecture provides a refined and comprehensive feature representation that enhances classification accuracy beyond traditional single-spectrogram approaches.

## 2. Literature Review

Deep learning for music informatics has demonstrated that convolutional neural networks (CNNs) can effectively learn features directly from audio data, advancing automatic feature extraction and classification [9]. The integration of Deep Learning Networks [10] and spectrogram features has proven to be a robust approach for audio classification, even in noise conditions [11]. Spectrograms, which represent audio signals visually, allow CNNs to learn intricate patterns and features from the time-frequency domain [12–14].

Transforming audio samples into spectrograms provides an effective advantage for neural network training, particularly with convolutional neural networks (CNNs). This approach not only optimizes the data for network input but also reduces the input size, offering a more efficient alternative to raw digital sample-based methods [15–18]. Research has demonstrated the effectiveness of CNNs in classifying musical instruments by employing different spectrogram methods. For example, one study [19] reports an 80% accuracy using the STFT (short-time Fourier transform) spectrogram. Additionally, approaches leveraging MFCCs (Mel-Frequency Cepstral Coefficients) have also been applied to instrument recognition, achieving notable accuracy level [20–22]. The authors of [23] introduced a method to categorize string instruments through the use of Chroma-based features.

Also, some research [24] proposed a multi-spectrogram encoder–decoder framework that utilizes different types of spectrograms to improve acoustic scene classification. Their approach highlights how integrating diverse spectral features can enhance the robustness of audio classification models.

Various studies [25,26] have explored different spectrogram types, such as STFT, Log-Mel, and MFCC, to capture diverse audio characteristics. Specifically, for musical instrument classification in recorded polyphonic music, the work [27] accurately classifies the NSynth dataset [28] with good performance. The IRMAS [29] dataset's work [30] achieved a 0.79 precision. The Open-Mic [31] dataset has been used more, with one work [32] achieving a 0.843 mean average precision (mAP). Another work [33] achieved a 0.852 mAP, and a benchmark work [34] achieved a 0.855 mAP.

## 3. Method

### 3.1. Theoretical Background

In this study, we investigate how scaled multi-spectrogram inputs affect the performance of a hierarchical residual attention network in musical instrument recognition. According to the Universal Approximation Theorem [35], a neural network can approximate any continuous function, as shown in (1):

$$Y = \sigma(w \cdot x + b) \tag{1}$$

where $Y$ is the output, $w$ is the weight, $x$ is the input, and $b$ is the bias. Here, $\sigma$ represents an activation function, such as the sigmoid or tanh, which limits (or squashes) the output into a specific range and introduces non-linearity to the model.

Thus, the spectrogram-based musical instrument classification can be expressed as

$$instrument = \sigma(w \cdot \text{spectrogram}(x) + b) \tag{2}$$

### 3.2. Hypothesis

Building on the theorem, we preprocess our spectrogram inputs by applying distinct scaling factors to each spectrogram component before feeding them into the model. This preprocessing step is formalized in (3):

$$Y = \sigma(w(S_1 \cdot LogMel(x) + S_2 \cdot Chroma(x) + S_3 \cdot SpectralContrast(x) \\ + S_4 \cdot Tonnetz(x)) + b) \tag{3}$$

where $Y$ is the output and $w$ represents the overall weight applied to the combined spectrograms, $(S_1, S_2, S_3,)$ and $(S_4)$ are the scale for the Log-Mel, Chroma, Spectral contrast, and Tonnetz spectrogram features, respectively, and $(x)$ represents the input audio signal.

The hypothesis (3) behind this approach is that by combining multiple spectrograms with assigned scales, this preprocessing method can provide a richer set of learnable features, improving the performance of the model. This scaled combination of spectrogram inputs is expected to better capture the nuances in audio signals that single-spectrogram methods might miss, thereby enhancing overall classification accuracy.

### 3.3. Data Pre-Processing

In this experiment, we use the OpenMIC-2018 dataset [31], an open-source, multilabel music instrument annotated database. This dataset does not control for specific physical parameters, such as subglottal pressure, vocal fold geometry, or recording equipment settings, as the dataset was designed to facilitate general audio analysis rather than studies requiring tightly controlled experimental conditions.

The dataset contains over 20,000 ten-second audio clips, each annotated with multiple instrument labels, providing a diverse and extensive collection of musical instrument samples for analysis. The dataset includes 20 different musical instruments, and all recordings are real audio recordings capturing the authentic characteristics of various instruments in natural settings.

We used the OpenMIC-2018 default train–test split, which includes 14,914 training samples and 5084 testing samples. During the training process, we further divided the training set by defining a validation split of 20%, with a random shuffle applied to ensure balanced representation across classes. This setup allows for consistent evaluation of model performance on unseen data while using a portion of the training data to monitor and fine-tune the model during training.

Figure 1 presents various audio feature representations extracted from a musical instrument sample, along with the effects of combining these features at different scales. Panel (a) shows the individual spectrogram types: the Log-Mel spectrogram has 128 Mel frequency bins, the Chroma feature is divided into 12 pitch classes, the Spectral contrast feature has 7 frequency bands, and the Tonnetz feature consists of 6 bins representing harmonic intervals. This is the default input dataset configuration used in the experiment, with no scaling adjustments applied to the spectrograms.

Using OpenCV [36], we resize the Chroma, Spectral contrast, and Tonnetz (CST) features to match the Log-Mel spectrogram's dimensions, as Log-Mel has shown superior performance in musical instrument classification research. In panel (b), all CST features are scaled to match the Log-Mel's size exactly. Panels (c), (d), and (e) illustrate alternative scaling approaches where CST features are resized to 1/4, 1/2, and 3/4 of the Log-Mel size, respectively. This scaling comparison explores the impact of different size ratios on feature representation and classification performance.

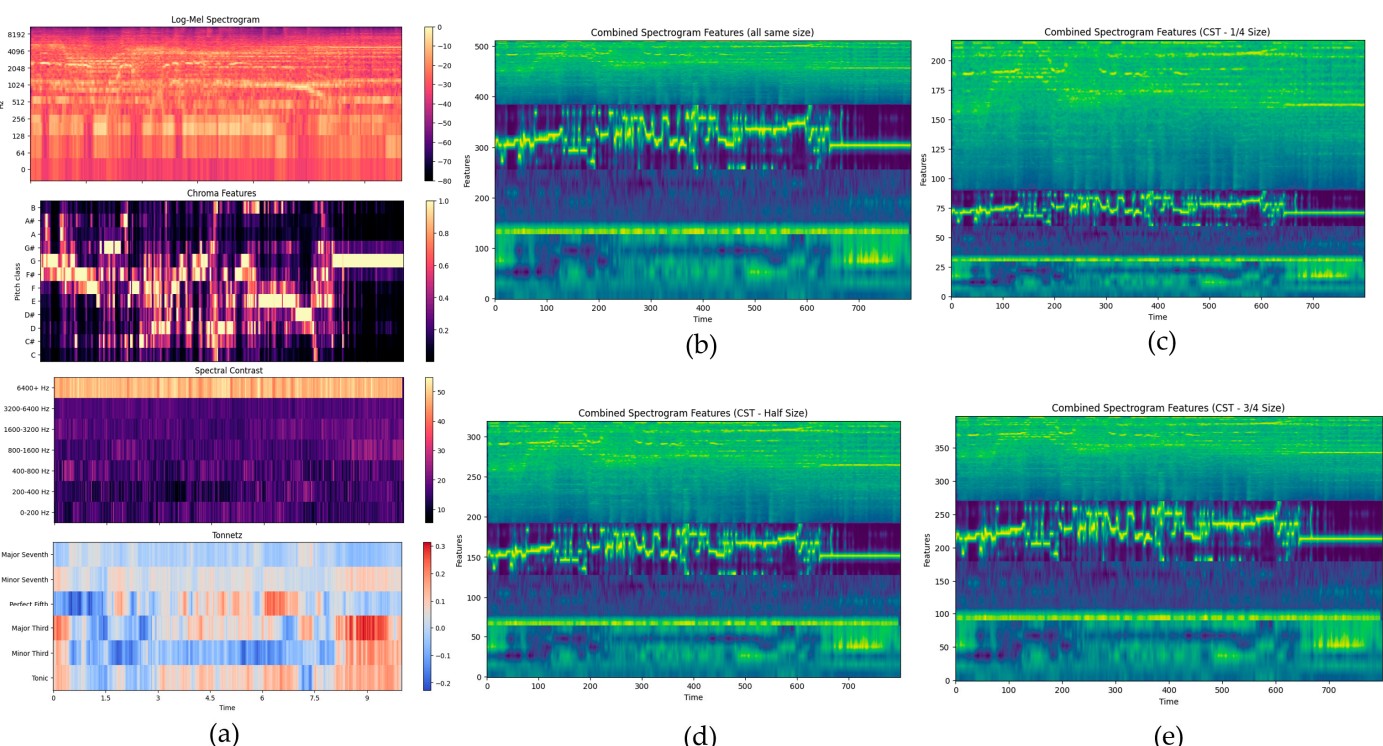

**Figure 1.** (**a**) From top to bottom: Log-Mel Spectrogram shows frequency (Hz) over time, where brighter regions represent higher intensity in decibels (dB). Chroma Features capture pitch class profiles (C, C#, D, D#, etc.), with "#" indicating a sharp note in musical notation. Spectral Contrast represents the difference in intensity between spectral peaks and valleys across frequency bands (e.g., 6400+ Hz to 200–400 Hz), providing complementary information to the Log-Mel Spectrogram. Tonnetz encodes harmonic relationships such as tonic, minor third, major third, perfect fifth, minor seventh, and major seventh, which represent musical intervals over time. (**b**) Combined Spectrogram Features (all same size), (**c**) Combined Spectrogram Features (CST—1/4 Size), (**d**) Combined Spectrogram Features (CST—Half Size), and (**e**) Combined Spectrogram Features (CST—3/4 Size) illustrate normalized feature values scaled between 0 and 1, where brighter colors indicate values closer to 1, and darker colors indicate values closer to 0. This normalization ensures consistent visualization despite differences in the original scales (e.g., frequency reaching 8192 Hz in Log-Mel, compared to the Tonnetz scale with a maximum value of 6 (Major Seventh)).

### 3.4. Convolutional Neural Network Structure

The neural network model used here (Figure 2) classifies musical instruments using spectrogram inputs. It consists of three residual blocks with 32, 64, and 128 filters, respectively, each followed by MaxPooling to reduce dimensions. Attention mechanisms are applied at multiple stages: Early Attention after the first block, Mid Attention after the second, and Late Attention after the third, along with Channel and Coordinate Attention to emphasize critical features. The output is flattened, passed through a dense layer with 512 units and ReLU activation, followed by a 40% dropout layer, and finally, a sigmoid activation layer for multi-label classification. The model uses binary cross-entropy loss, Adam optimizer, and accuracy as the metric.

The combined use of Coordinate [7] and Channel Attention [37] mechanisms enable a more flexible, targeted, and thorough analysis of the spectrogram inputs. By adjusting focus along both spatial and channel dimensions, the model can learn hierarchical representations that capture combined spectrogram features of musical instruments with greater specificity. This multi-stage attention application supports improved generalization, even in challenging and noisy audio conditions. This architecture effectively combines residual

connections, attention mechanisms, and dense layers to achieve robust feature extraction and classification performance for musical instrument recognition tasks.

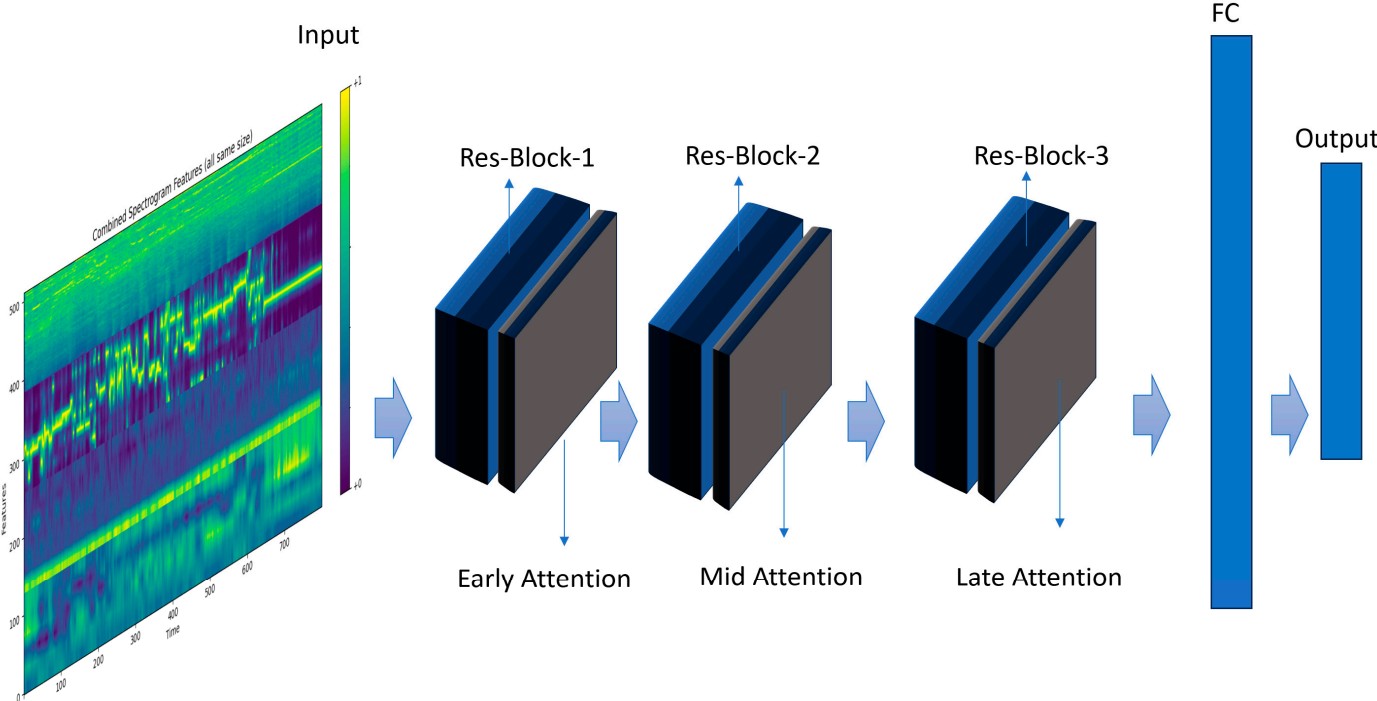

**Figure 2.** Overview of the neural network architecture used for musical instrument classification. The input is an example sample of combined spectrogram features (CST) with all components scaled to the same size and normalized to a range between 0 and 1, where brighter colors indicate values closer to 1. The model includes three residual blocks (Res-Block-1, Res-Block-2, Res-Block-3), shaded in blue and dark blue represent operations such as convolution, batch normalization, and activation functions. Each residual block is followed by a coordinate attention mechanism (Early Attention, Mid Attention, Late Attention) to enhance feature representation. After the three residual blocks and attention layers, the final fully connected (FC) layer performs the classification to generate the output.

We trained the CNN model for up to 500 epochs with a batch size of 32, using early stopping with a patience of 100 epochs to prevent overfitting and retain the best model weights. A learning rate reduction was applied with a patience of 50 epochs and a decay factor of 0.1 to fine-tune the learning process. Training also incorporated sample weights to address class imbalances in the dataset. Detailed parameters are listed in Data Availability Statement (DAS) Section.

## 4. Results

### 4.1. Benchmark Comparison

Figure 3 presents a comparison of mean average precision (mAP) achieved by different methods on the Open-MIC dataset. The x-axis lists the methods, while the y-axis indicates the mAP values. The graph includes benchmark methods and our proposed methods and highlights our best model. The benchmark methods serve as a reference point, showing the progression in performance over the years, including the Baseline [31], PaSST [32], EAsT-KD + PaSST [33], and DyMN-L [34]. Our methods include various configurations of combining spectrogram features and attention mechanisms. Specifically, the "Single Log-Mel" approach, "Log-Mel CST Combined Spectrogram", and "Log-Mel CST with Attention Layer" configurations explore the impact of different spectrogram features on model performance. Additionally, we experimented with magnifying the spectrogram features to different extents: "Magnified 1/4 Size", "Magnified 1/2 Size", "Magnified 3/4 Size", and "Magnified Full Size".

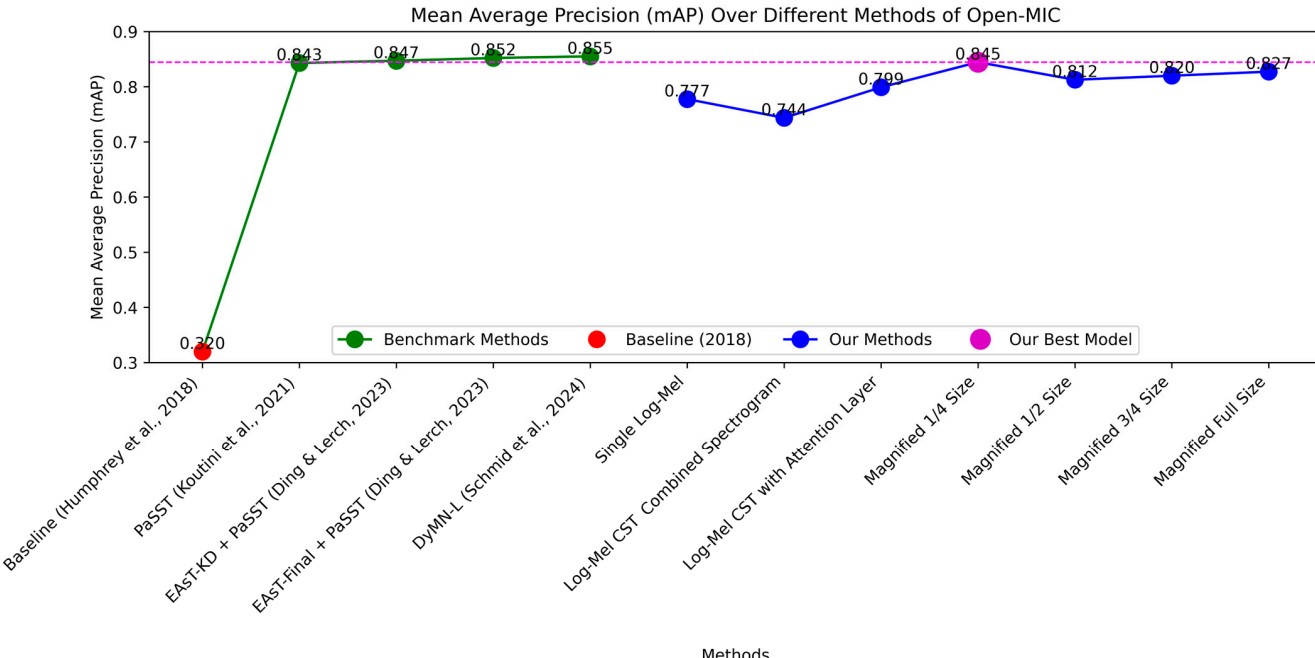

**Figure 3.** Mean average precision (mAP) comparison across various methods on the Open-MIC dataset [31–34].

Among these, the "Magnified 1/4 Size" model achieved a mAP of 0.8445, demonstrating a performance close to the leading benchmark methods. This model's success highlights the importance of carefully scaling the spectrogram features and incorporating attention mechanisms to enhance the model's focus on the most informative parts of the input data. The magenta marker and horizontal magenta line on the graph emphasize the noteworthy performance of this model, illustrating the potential effectiveness of our approach in musical instrument recognition tasks.

### 4.2. Evaluation Metrics Comparison Among Each Scaled Multi-Spectrogram Settings

Figure 4 presents a comprehensive comparison of the precision, recall, and F1 score metrics for various instrument recognition models using different configurations of spectrogram features. The x-axis represents the different musical instruments, while the y-axis indicates the metric values.

The models compared include configurations such as Log-Mel 128 with Original CST Sizes (Chroma = 12, Spectral Contrast = 7, Tonnetz = 6), Log-Mel 128 with CST Magnified to 1/4 Size (Chroma = 32, Spectral Contrast = 32, Tonnetz = 32), Log-Mel 128 with CST Magnified to 1/2 Size (Chroma = 64, Spectral Contrast = 64, Tonnetz = 64), Log-Mel 128 with CST Magnified to 3/4 Size (Chroma = 96, Spectral Contrast = 96, Tonnetz = 96), and Log-Mel 128 with CST Magnified to Full Size (Chroma = 128, Spectral Contrast = 128, Tonnetz = 128). Each sub-plot within the figure illustrates a specific metric comparison, with the top plot showing precision comparison, the middle plot showing recall comparison, and the bottom plot showing F1 score comparison.

In Table 1, the mean precision remains stable across CST sizes, ranging from 0.812 to 0.845, with a consistent standard deviation of 0.12, indicating steady model performance in detecting instruments. Recall shows a slight variation, with mean values between 0.16 and 0.18 and a higher standard deviation for 1/4 and 3/4 sizes (0.14) compared to 1/2 (0.11), indicating variability in detecting specific instruments for certain configurations. The F1 score has a mean range of 0.25 to 0.27, with standard deviations from 0.15 to 0.18, highlighting some variability in balanced performance, especially at the 1/4 and full sizes.

Overall, Table 1 suggests that while the 1/4 and full sizes achieve slightly higher F1 scores, they also introduce more variability across instruments. This variability hints that

certain CST sizes may enhance performance for specific instruments but could reduce consistency across the broader set, indicating a trade-off between targeted accuracy and general stability across CST configurations.

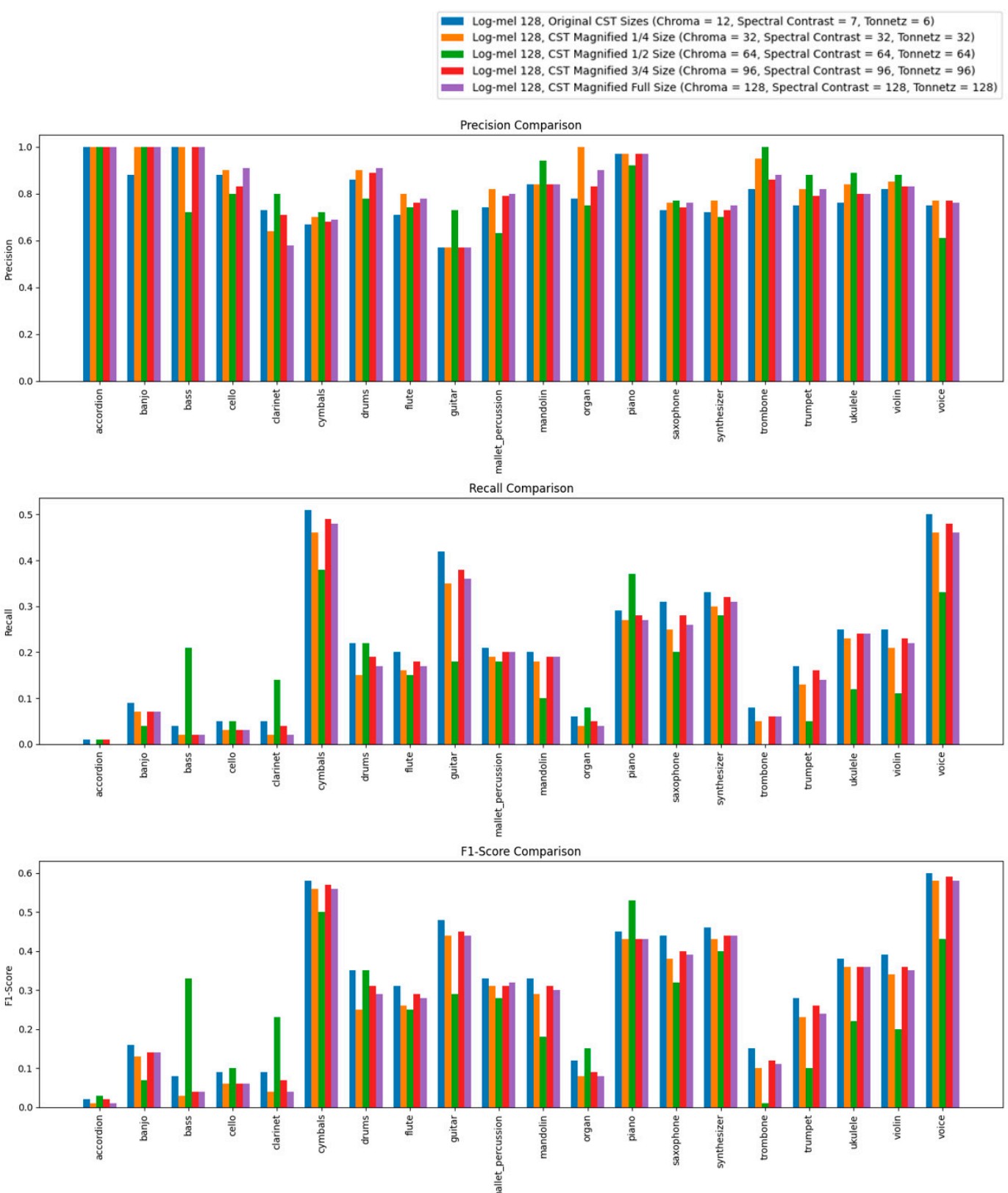

**Figure 4.** Precision, recall, and F1 score comparisons for different spectrogram scaled sizes.

**Table 1.** Mean and standard deviation of precision, recall, and F1 score for different CST sizes, highlighting performance stability and variability across instruments.

| Size Setting | Precision | Std. Precision | Mean Recall | Std. Recall | Mean F1 | Std. F1 |
|---|---|---|---|---|---|---|
| ¼ CST Size | 0.845 | 0.12 | 0.18 | 0.14 | 0.27 | 0.18 |
| ½ CST Size | 0.812 | 0.12 | 0.16 | 0.11 | 0.25 | 0.15 |
| ¾ CST Size | 0.820 | 0.12 | 0.17 | 0.14 | 0.26 | 0.17 |
| Full Size | 0.827 | 0.12 | 0.17 | 0.14 | 0.26 | 0.17 |

OpenMIC-2018 is a multilabel [38] dataset where samples can contain multiple instruments; we use individual confusion matrices for each instrument to evaluate model performance. According to Figures 4 and 5, for instruments like accordion, banjo, bass, drums, guitar, marimba, piano, synthesizer, and trumpet, a high degree of precision is consistently maintained across all configurations, indicating effective differentiation with minimal false positives. However, for cello, clarinet, flute, mandolin, violin, and voice, precision varies, suggesting certain spectrogram features better reduce false positives. Notably, the "Log-Mel 128 CST Magnified 1/4 Size" configuration generally provides a balanced performance, capturing essential characteristics effectively. Instruments like cymbals, organs, saxophones, and trombones exhibit significant precision fluctuations, indicating overlapping features with other instruments, making accurate differentiation more challenging.

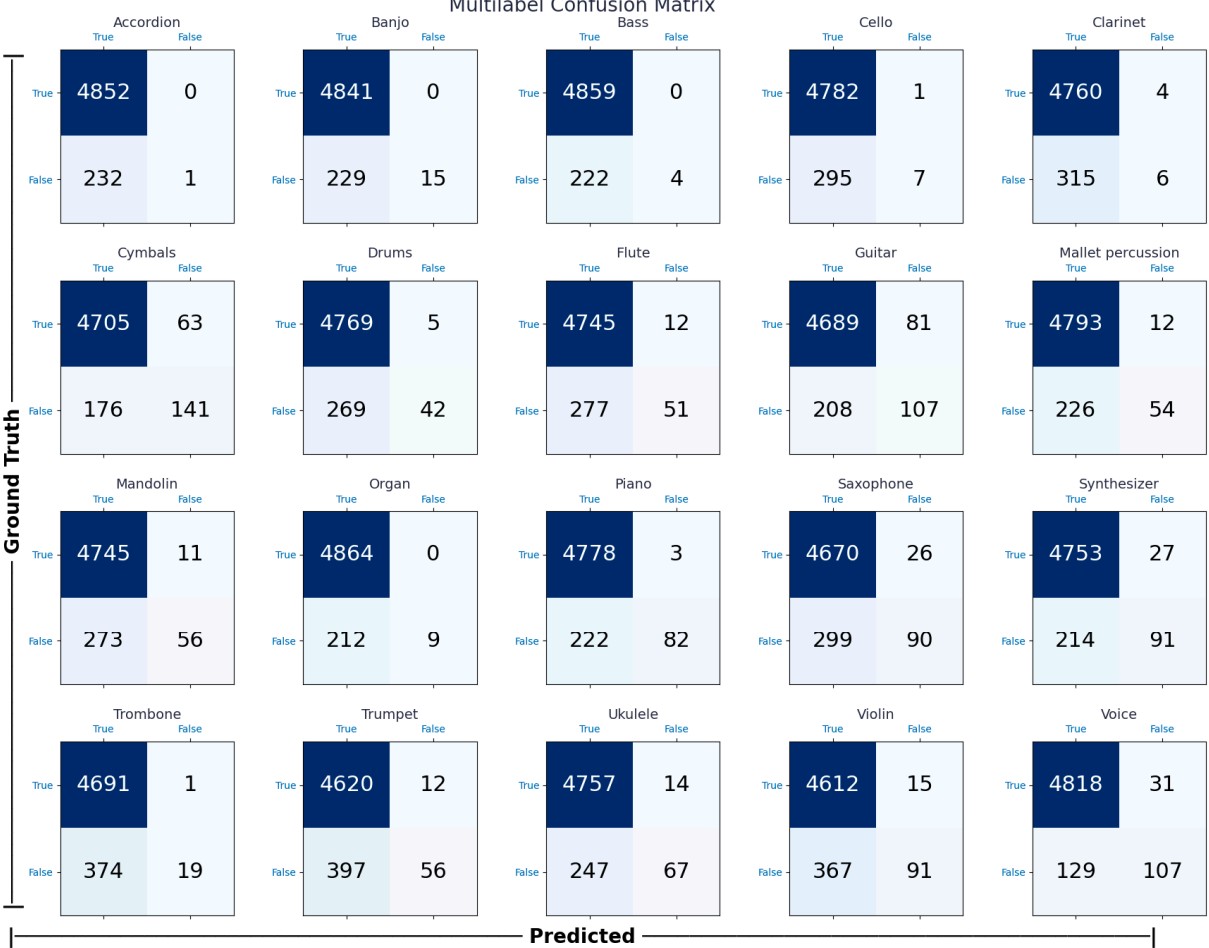

**Figure 5.** Multilabel confusion matrices for the best-performing model on each instrument. The rows represent the ground truth labels, while the columns indicate the predicted classifications. The matrices display true positives, true negatives, false positives, and false negatives for every instrument classification. Darker shades represent higher values, illustrating areas of strong classification accuracy and confusion points between certain instruments.

Recall comparison (Figure 4) reveals consistently high detection rates for accordion, banjo, bass, piano, synthesizer, trumpet, ukulele, violin, and voice, showcasing the model's effectiveness. Variability in recall for cello, clarinet, cymbals, flute, guitar, marimba, mandolin, and saxophone, with the "Log-Mel 128 CST Magnified Full Size" often achieving higher recall, suggests larger feature sizes capture more relevant characteristics. Lower and more variable recall rates for drums, organs, and trombones indicate that these instruments are less distinct or more challenging to detect accurately. High and consistent F1 scores for instruments like accordion, banjo, bass, drums, guitar, piano, synthesizer, and trumpet reflect a good balance between precision and recall. In contrast, cello, clarinet, cymbals, flute, mandolin, organ, saxophone, and trombone show fluctuating F1 scores, with the "Log-Mel 128 CST Magnified 1/4 Size" and "Log-Mel 128 CST Magnified Full Size" configurations often performing better. This indicates these configurations provide a better trade-off between detecting instruments and minimizing false predictions, while voice, violin, marimba, and ukulele show variability in F1 scores, suggesting room for optimization in feature size and attention mechanisms.

## 5. Discussion

The different configurations of CST feature sizes (original, 1/4, 1/2, 3/4, and full) highlight how scaling affects model performance. Smaller sizes generally provide a compact representation, leading to higher precision but potentially lower recall. Conversely, larger sizes capture more details, improving recall but possibly introducing more false positives. The "Magnified 1/4 Size" configuration often achieves a good balance, making it a preferred choice for general purposes. The variability in performance across different instruments suggests that certain instruments benefit more from specific feature configurations. For instance, instruments like drums and organs might require more sophisticated feature combinations or additional attention mechanisms to enhance detection accuracy. Incorporating attention mechanisms at various stages helps the model focus on the most relevant parts of the spectrogram, improving overall performance. The results indicate that these mechanisms are crucial, particularly for instruments with overlapping or subtle features. Further research could explore optimizing CST feature sizes and experimenting with other attention mechanisms. Additionally, applying the hierarchical residual attention network to other audio classification tasks could test its generalizability and effectiveness beyond musical instrument recognition.

### 5.1. Early Attention Layer Analysis

The first two rows in Figure 6 illustrate the early convolutional layers, which capture basic spectral and harmonic structures within the input spectrogram. At this stage, the feature maps display a high level of activity across the spectrogram, revealing that the network is primarily learning to detect general patterns such as fundamental frequencies and overtone series. These early Conv layers act as foundational filters, identifying broad regions of frequency and amplitude that are essential for distinguishing basic sound structures. The activity patterns suggest that the network is still focused on general sound characteristics, laying the groundwork for more detailed differentiation in later layers.

The third and fourth rows depict the height and width attention maps. In the height attention maps, the model emphasizes certain frequency bands, suggesting that it is beginning to recognize which harmonic regions contribute most to the trumpet and bass timbres. This selective focus highlights early attempts at isolating key harmonic frequencies. In the width attention maps, the network shows attention to temporal segments, indicating an emerging sensitivity to rhythmic or temporal consistency across the input sound. These attention maps reveal how the model begins to prioritize significant areas of the spectrogram, helping it capture the foundational time-frequency patterns.

Early Feature and Attention Maps of a Trumpet + Bass Sample

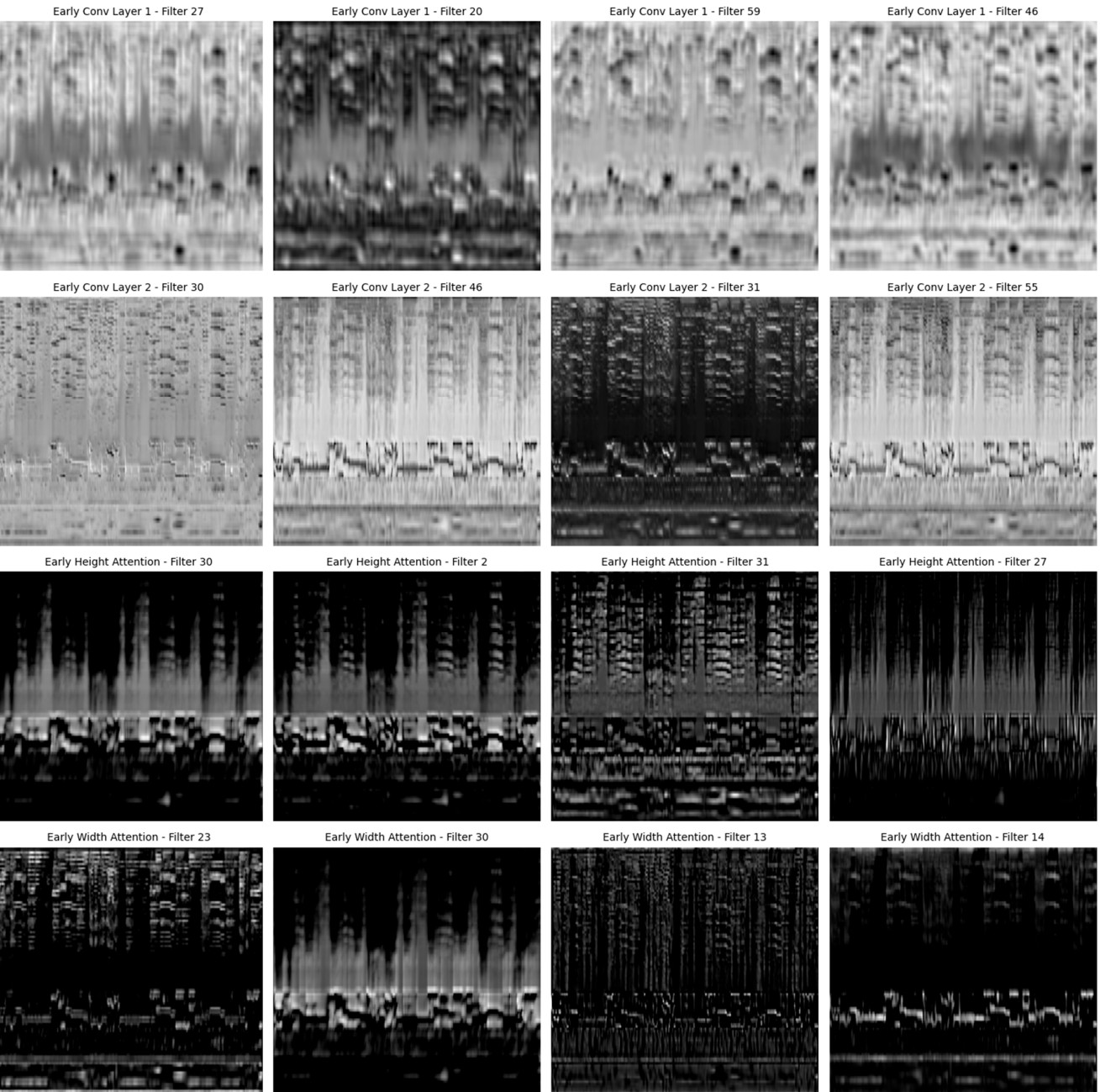

**Figure 6.** Early feature and attention maps for a trumpet and bass sample showing the initial layer outputs of the Log-Mel and scaled CST (Chroma, Spectral contrast, and Tonnetz) features. The maps highlight the fundamental frequency components and outline basic musical structures through a combination of Log-Mel's high-resolution representation and the CST features scaled to 1/4 of the Log-Mel size. The grayscale shades represent values in the feature maps and attention maps, where darker shades indicate lower values, and lighter shades indicate higher values, providing a visual representation of the extracted features.

## 5.2. Mid Attention Layer Analysis

The first two rows in Figure 7 show the mid-layer convolutional outputs, where the network's feature maps become more abstract and complex. Here, the model is no longer capturing only basic spectral patterns; instead, it begins to identify mid-level structures,

such as harmonic relationships and timbral qualities specific to each instrument. The feature maps display more intricate textures and organized clusters, reflecting the model's refined understanding of trumpet and bass sounds. These mid-level Conv layers capture more nuanced aspects of sound, helping the network to distinguish between instruments with increased specificity.

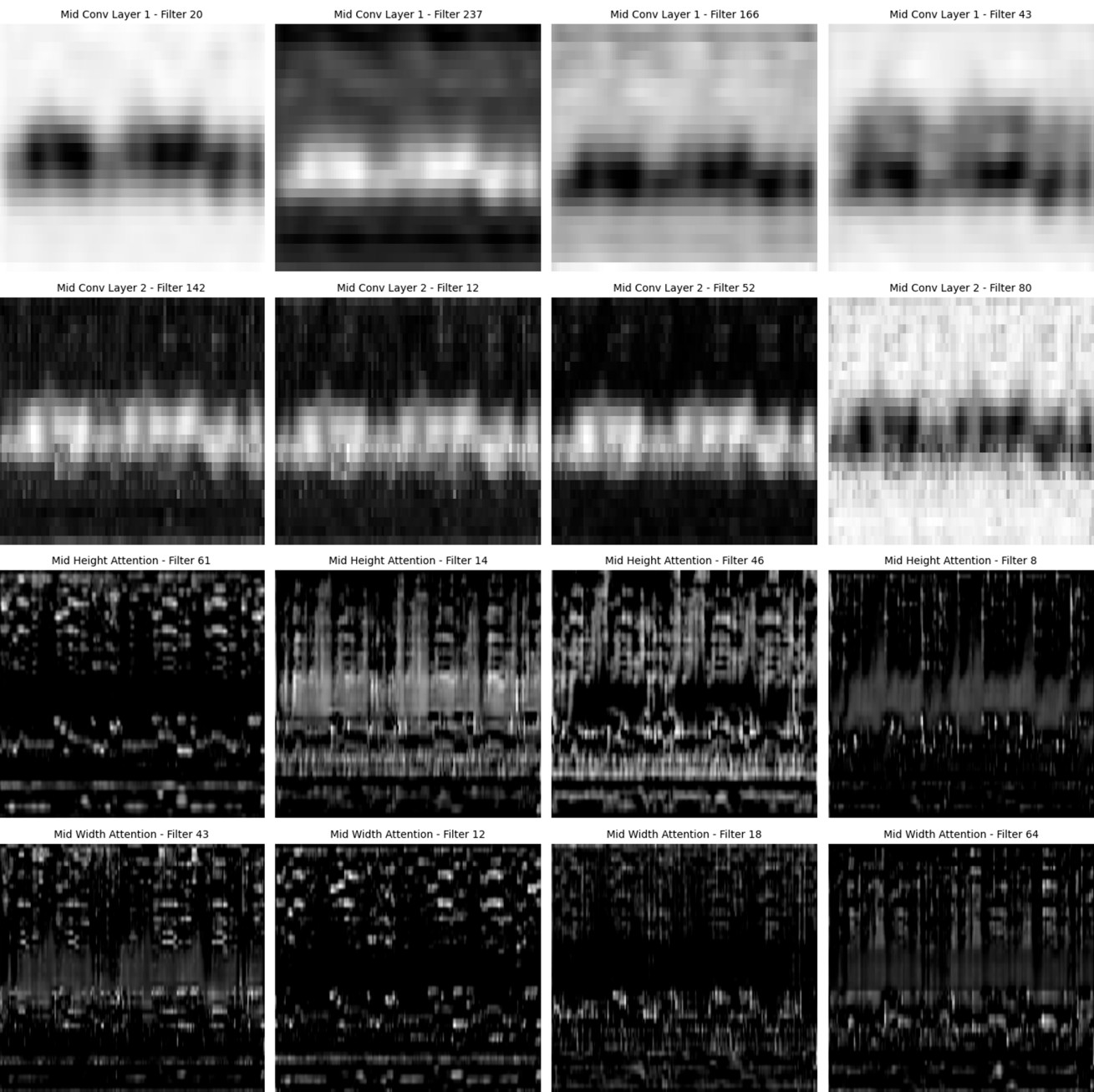

**Figure 7.** Mid-level feature and attention maps for a trumpet and bass sample, capturing intermediate representations with heightened detail through the combination of Log-Mel spectrogram and CST features at 1/4 scale. The mid layers reveal more refined frequency and harmonic structures, enhancing the model's ability to distinguish between complex timbral characteristics. The grayscale shades represent values in the feature maps and attention maps, where darker shades indicate lower values, and lighter shades indicate higher values, providing a visual representation of the extracted features.

The height and width attention maps in the mid layers, shown in the third and fourth rows, demonstrate a sharper and more targeted focus. The height attention maps now isolate specific harmonic frequencies more accurately, suggesting that the model has identified frequency bands that are especially characteristic of each instrument's tonal quality. In the width attention maps, the network's focus on temporal segments is also more refined, capturing rhythm and temporal dynamics with greater precision. This enhanced attention to both frequency and time dimensions enables the model to extract timbral and rhythmic cues that contribute to the mid-level understanding of each instrument.

### 5.3. Late Attention Layer Analysis

The first two rows in Figure 8 highlight the late Conv layers, where feature maps become highly selective and abstract. These layers capture distinct, high-level representations that embody the unique sound signatures of each instrument. The network has refined its focus to specific frequency and amplitude regions, isolating instrument-specific characteristics like the spectral textures unique to trumpet and bass. The sparse yet sharply defined patterns in these feature maps indicate that the model has distilled the input into concentrated features necessary for final classification. This layer's focus on highly abstract, distinguishing features reflects the network's advanced ability to identify each instrument's unique sound signature.

In the height and width attention maps of the late layers (third and fourth rows), we see a strong, focused emphasis on select frequency and time segments. The height attention maps isolate frequency regions that contribute most to the tonal identities of trumpet and bass, likely capturing Chroma components that characterize the harmonic profile of each instrument. The width attention maps display precise temporal focus, essential for capturing articulation and rhythmic qualities that differentiate the instruments. This advanced focus on time and frequency allows the model to highlight key sound features, capturing the unique temporal dynamics and harmonic content crucial for final instrument classification.

In the final row, the channel attention maps show the model's ability to focus on abstracted, high-level channel features. These maps reveal selective emphasis on specific channels, which are likely to correspond to spectral regions that capture the most informative cues for instrument identification. By isolating and concentrating on these channels, the network leverages the most relevant aspects of the combined Log-Mel and scaled CST features. This channel attention helps synthesize frequency and temporal information into a cohesive representation, enabling the model to accurately distinguish between complex instrumental sounds, completing the classification process with a robust, high-level understanding of each instrument.

Late Feature and Attention Maps of a Trumpet + Bass Sample

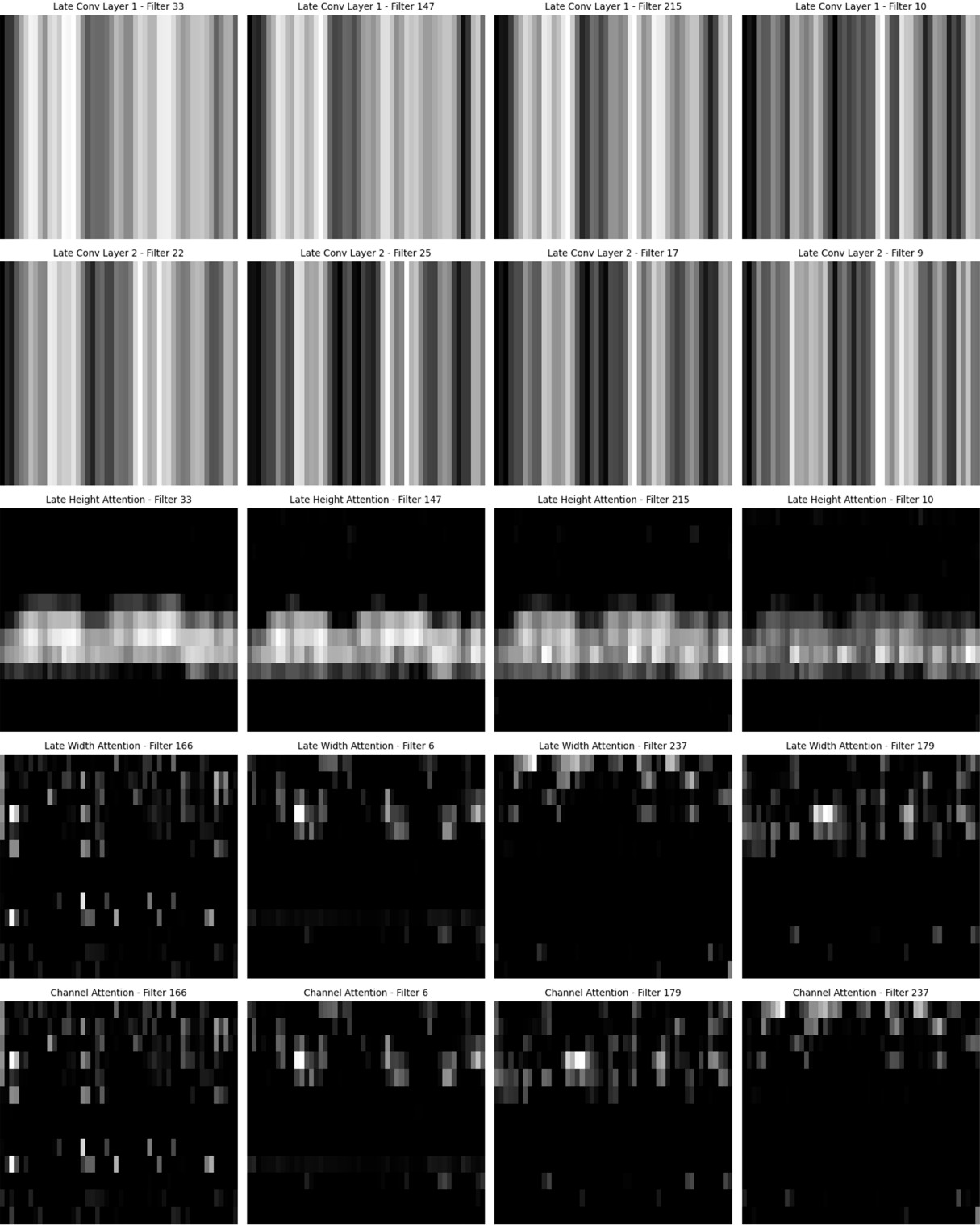

**Figure 8.** Late feature and attention maps for a trumpet and bass sample, illustrating high-level abstracted representations formed by deeper layers. The Log-Mel spectrogram and scaled CST features at 1/4 size contribute to an enriched final classification layer, enhancing the recognition of instrument-specific patterns through focused, high-contrast attention maps. The grayscale shades represent values in the feature maps and attention maps, where darker shades indicate lower values, and lighter shades indicate higher values, providing a visual representation of the extracted features.

## 6. Conclusions

In this study, we presented a hierarchical residual attention network for musical instrument recognition using scaled multi-spectrogram features. By combining the Log-Mel feature with CST features (Chroma, Spectral contrast, and Tonnetz), our model captures a more comprehensive representation of audio signals. The attention mechanisms, applied at various stages of the network, allow the model to focus on the most relevant parts of the combined spectrogram, improving classification performance.

Our experiments demonstrated that the "Magnified 1/4 Size" configuration achieved the best balance of precision, recall, and F1 score, highlighting the effectiveness of scaling spectrogram features to enhance model performance. The use of hierarchical residual connections and attention mechanisms significantly improved the model's ability to classify instruments accurately, even those with subtle or overlapping features.

## 7. Limitations

This proposed approach improves instrument recognition by leveraging scaled multi-spectrogram features and hierarchical attention mechanisms; however, it has several limitations. Firstly, the method relies on manually selected spectrogram scaling configurations, which may not be optimal for all instruments. This reliance on fixed scaling parameters can lead to inconsistent performance across various instrument types, as certain scales might emphasize features relevant to some instruments while underrepresenting others. Additionally, the model's performance heavily depends on the quality and diversity of the training dataset. Limited training samples for less common instruments could result in reduced accuracy for these classes, as the model may not generalize as effectively to instruments with fewer examples.

Another limitation is the increased computational complexity introduced by the layered attention mechanisms. While the hierarchical attention layers enhance the model's ability to focus on relevant frequency and time features, they also make the approach computationally intensive. This complexity could restrict the model's usability in real-time applications or deployment on devices with limited processing power, such as mobile or embedded systems. Furthermore, the proposed method has not been extensively tested across different audio environments, which may impact its robustness in noisy or acoustic varied settings. Future work could address these challenges by investigating adaptive scaling techniques, optimizing model efficiency, and testing the approach under diverse real-world conditions to enhance its applicability and reliability.

## 8. Future Work

We recognize that the performance improvements achieved here have room for further enhancement. Nonetheless, the use of scaled multi-spectrogram features creates opportunities for further advancements in audio recognition models.

Thus, future work will explore optimizing scaling parameters by making them learnable within the network, enabling the model to adjust each feature's representation during training for enhanced accuracy.

Additionally, we can investigate multi-channel spectrogram inputs where each channel applies a different scale or introduce specific attention layers per spectrogram type to improve feature distinction. Another direction involves adaptive attention mechanisms, such as Vision Transformer [39], which can adapt feature weighting based on both spectral and temporal patterns. These strategies could strengthen the model's focus and improve its ability to capture complex audio features, addressing some limitations observed in the current approach.

**Author Contributions:** Conceptualization, A.N.; Methodology, A.N.; Software, R.C.; Validation, A.G.; Writing—original draft, R.C.; Writing—review & editing, A.G. and A.N.; Supervision, A.G. and A.N. All authors have read and agreed to the published version of the manuscript.

**Funding:** This research received no external funding.

**Institutional Review Board Statement:** Not applicable.

**Informed Consent Statement:** Not applicable.

**Data Availability Statement:** The data presented in this study are openly available. The OpenMIC-2018 dataset, used in this study, can be accessed at https://zenodo.org/record/1432913 (accessed on 18 November 2024). The code used for experimenting on this dataset is publicly available at https://github.com/fireHedgehog/music-intrument-OvA-model/tree/main/open-mic (accessed on 18 November 2024).

**Conflicts of Interest:** The authors declare no conflict of interest.

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
