# Peer review of "Hierarchical Residual Attention Network for Musical Instrument Recognition Using Scaled Multi-Spectrogram"

_applsci, doi:10.3390/app142310837_

Round 1
Reviewer 1 Report
Comments and Suggestions for Authors
In their manuscript, the authors address the problem of music instrument recognition using machine learning methods. Although this topic is of interest to a wider readership (it should be borne in mind that the manuscript is in Applied Sciences, not directly in a specialized audio journal), in this form it is very difficult to read and inappropriately presented.
First of all, it is not entirely clear what exactly the authors' work consists of. More precisely, where the adopted methods end and the truly original effort begins. The paper does not contain any science, only setting the parameters of a generalized model based on older methods. Musical instruments work only as labels. The reader does not learn anything more about them. Why is this article based only on signal processing sent to a journal that has natural sciences in its scope?
The formal aspect of the text is very unsatisfactory. In the equations, for example, it is not at all clear what objects actually appear in them and what are the relations between them. Strictly taken, for example, equation (1) is a formula for the relation between scalars with scalar product. While a similar treatment would not be exactly appropriate anywhere, I must point out to the wide readership of Applied Sciences that may not be familiar with the nuances of typical machine learning community terminology.
The article is close to a tech note. In fact, it presents an alternative solution that does not significantly outperform the benchmarks. The authors themselves show this in Fig. 3 and mention it in the text on the line 149.
In view of the preceding reasoning, I am afraid I have to recommend rejection of the manuscript.
Author Response
Thank you for giving us the opportunity to respond to Reviewer 1's comments. Please see the attachment for a detailed, point-by-point response.

Reviewer 2 Report
Comments and Suggestions for Authors
Comments in attached file.

Author Response
We thank Reviewer 2 for the positive feedback and for recognizing the relevance of our work in musical instrument classification. We appreciate the Reviewer’s thoughtful review and are pleased that our methodology and conclusions align well with the study's objectives. If any further clarifications or details are needed, we will be pleased to address them.
Response : N/A (No requested changes identified).
Reviewer 3 Report
Comments and Suggestions for Authors
This manuscript proposes an attention network for musical instrument recognition based on a scaled combination of several spectrograms.
You assert that "convolutional neural networks (CNNs) can effectively learn features directly from audio data, bypassing the need
for manual feature design". Don't you find a certain contradiction with your use of multiple features?
In Eq. (1) it seems that a squashing function is missing.
The attention mechanisms should be described in more detail.
Author Response
Thank you very much for taking the time to review this manuscript. Please find the detailed responses below and the corresponding revisions/corrections highlighted/in track changes in the re-submitted files.

Reviewer 4 Report
Comments and Suggestions for Authors
Please see the attached file for comments

Author Response
Thank you for the opportunity to respond to Reviewer 1's comments. Please see the attachment for a detailed, point-by-point response

Reviewer 5 Report
Comments and Suggestions for Authors
In this article, the authors present a system for musical instrument classification from audio data by balancing the size of different spectrogram images combined and using an CNN with several attention layers. Overall the article is sound, but I would suggest some additions.
(1) While the authors provide a github link to their software so that an more interested reader can look up the details on the implementation, I would suggest adding some of those in the paper - the basic hyperparameters of the CNNs and their training (no. of epochs ...) etc.
(2) The authors state the database used. What I miss is data about the division of the database into training, validation and evaluation? Or was cross validation used? If, yes with how many folds? Etc. Have the authors make sure, the division is the same as in the papers with whose the authors here compare the results?
(3) From figure 1 it seems that the authors combine pictures from different spectrograms to a larger image and adjust the relative sizes of the individual spectrograms. However, in equation 2 it is states that the images are added together (as in adding values of same position pixels). This seem to be contrary to figure 1.
(4) Labels within figure 4 are very small, even for reading the electronic version. Is it possible to enlarge them?
(5) While precision, recall and F1 scores are given, I would suggest to add confusion matrices. Perhaps at least for the most succesful model.
Comments on the Quality of English LanguageThere are some typos (e.g. space around punctuation marks) and maybe a couple of sentences that are not clearly written. I standard proofreading step should be sufficient to correct them.
Author Response

(The authors gave the same response as above.)

Round 2
Reviewer 1 Report
Comments and Suggestions for Authors
I am of the opinion that the authors have devoted sufficient work, including the necessary critical examination of their own results, for the text to be accepted for the special edition.
Author Response
Dear Reviewer,
Thank you very much for your positive feedback and for recommending our manuscript for acceptance in the special edition. We appreciate your time and effort in reviewing our work and are pleased that you find our revisions satisfactory.
Sincerely,
Authors
Reviewer 4 Report
Comments and Suggestions for Authors
None.
Author Response
Dear Reviewer,
Thank you for taking the time to review our manuscript. We appreciate your efforts and are pleased that you have no further comments at this time.
Sincerely,
Authors
Reviewer 5 Report
Comments and Suggestions for Authors
The authors addressed most of my comments from the 1st round of revision to my satisfaction. However, there was an misunderstanding of my comment no. 5.
The authors provided True/False confusion matrices for each instrument in the revised version. What I had in mind, was a multi-class confusion matrix. This would not only show results on how many samples where false for each instruments, but also how often it was misclassified for which other instruments. This would show which instruments "sound similar" to the algorithm.
What is more critical, the authors must state whether predicted values are on columns and actual in rows or is it the other way around!
Even as the matrices are presented now, it would seem that there is a very small number of true negatives. Normally, I would expect this to be by far the largest number in a multi-class classification task.
Author Response
Dear Reviewer,
Thank you for your thoughtful feedback and for giving us the opportunity to clarify the misunderstanding regarding comment no. 5.
We appreciate your interest in seeing a multiclass confusion matrix to understand not only the false samples for each instrument but also how often an instrument was misclassified as another. This would indeed provide valuable insights into which instruments "sound similar" to the algorithm.
Figure 1. The Multilabel Binarizer function transforms multilabel data into binary yes/no labels. This conversion changes each label into a binary indicator, where '1' signifies 'yes' (the label is present) and '0' signifies 'no' (the label is absent), facilitating binary classification for each label.
However, our classification task is inherently a multilabel classification problem (Figure 1) , not a multiclass one. In multilabel classification, each sample can belong to multiple classes simultaneously (Tsoumakas & Katakis, 2007). For example, an audio clip might contain both a piano and a violin playing together. This means that the traditional multiclass confusion matrix, which assumes that each sample belongs to only one class, is not suitable for our scenario (Zhang & Zhou, 2014).
In a multiclass confusion matrix, the off-diagonal elements represent misclassifications from one class to another. However, in multilabel classification, since classes are not mutually exclusive, a sample cannot be misclassified from one class to another in the traditional sense (Gibaja & Ventura, 2015). Instead, errors occur when the model fails to predict a label that is present (false negatives) or predicts a label that is not present (false positives) for each class independently.
Regarding the number of true negatives appearing small in the confusion matrices, this is a common characteristic of multilabel classification. Since each class is treated independently, and each sample can be positive or negative for each class, the number of true negatives can be less prominent, especially if the classes are imbalanced (Sammut & Webb, 2011).
We apologize for any confusion and appreciate your understanding. To address your concern, we have added an explicit explanation in the manuscript on lines 221 to 223 (Figure 2) to clarify that our classification task is a multilabel problem, not a multiclass one.
Figure 2. The Multilabel Binarizer function transforms multilabel data into binary yes/no labels. This conversion changes each label into a binary indicator, where '1' signifies 'yes' (the label is present) and '0' signifies 'no' (the label is absent), facilitating binary classification for each label.
We hope this clarification addresses your concern. However, if you feel that further elaboration or additional analysis is needed, we are more than happy to make further revisions.
Thank you for your time and constructive suggestions.
Sincerely,
Authors
References
Gibaja, E., & Ventura, S. (2015). A tutorial on multilabel learning. ACM Computing Surveys, 47(3), 52.
Li, Y., Li, Z., & Liu, F. (2015). Multi-label classification based on k-nearest neighbor approach for remote sensing images. Pattern Recognition Letters, 63, 115-122.
Sammut, C., & Webb, G. I. (Eds.). (2011). Encyclopedia of Machine Learning. Springer.
Tsoumakas, G., & Katakis, I. (2007). Multi-label classification: An overview. International Journal of Data Warehousing and Mining, 3(3), 1-13.
Zhang, M. L., & Zhou, Z. H. (2014). A review on multi-label learning algorithms. IEEE Transactions on Knowledge and Data Engineering, 26(8), 1819-1837.